# Effects of Nutrient Addition on *Pedicularis kansuensis* Invasion of Alpine Grassland

**Haining Li** [1], **Yanming Gong** [2,3], **Fei Fang** [4], **Kaihui Li** [1,2,3,*] **and Yanyan Liu** [2,3,*]

[1]  Xinjiang Laboratory of Grassland Resources and Ecology, College of Grassland Science,
    Xinjiang Agricultural University, Urumqi 830052, China
[2]  Xinjiang Institute of Ecology and Geography, Chinese Academy of Sciences, Urumqi 830011, China
[3]  Bayanbulak Steppe Ecosystem Research Station, Chinese Academy of Sciences, Bayanbulak 841314, China
[4]  Department of Ecology and Environment of Xinjiang Uygur Autonomous Region, Urumqi 830063, China
[*]  Correspondence: likh@ms.xjb.ac.cn (K.L.); liuyany@ms.xjb.ac.cn (Y.L.)

**Abstract:** In order to study the changes in invasive plant population characteristics under different nutrient addition treatments, this study used the native invasive species *Pedicularis kansuensis*, which is spreading in the Bayabulak alpine grassland, as the research object and conducted two consecutive years of field studies in which nutrients were added to plots. Changes in the *P. kansuensis* population's invasive characteristics were monitored in 2020 and 2021 in four different nutrient-addition treatments, namely no-nutrients (control), low-nitrogen, high-nitrogen, and phosphorus treatments. The result showed that (1) nutrient addition had significant effects on *P. kansuensis* height and root/shoot ratio ($p < 0.05$); the time effect had significant effects on *P. kansuensis* height, coverage, abundance, aboveground biomass, and belowground biomass ($p < 0.01$), and the interaction between nutrient addition and time had a significant effect on *P. kansuensis* height ($p < 0.01$). (2) Nitrogen addition effectively inhibited the growth and the development of *P. kansuensis*, especially under high-nitrogen conditions in the second growing season, where the effect of height (2.50 cm), coverage (0.13%), richness (3 strains), aboveground biomass (0.21 g m$^{-2}$), and belowground biomass (0.03 g m$^{-2}$) was significant, with the *P. kansuensis* population almost disappearing by the end of the trial. (3) Phosphorus addition had no significant effect on the *P. kansuensis* population's invasive characteristics. These results indicate that higher nitrogen addition could effectively slow the invasion of the *P. kansuensis* population, and the findings of this study could provide certain baseline data and scientific guidance for the effective control of the *P. kansuensis* invasion of the Bayabulak alpine grassland in the future as well as identify certain theoretical bases for the effect of nutrient addition on invasive plants overall.

**Keywords:** nutrient addition; population characteristics; *Pedicularis kansuensis*; invasive plants

## 1. Introduction

Global warming, precipitation change, and nitrogen deposition increase are significantly affecting the balance and stability of ecosystems [1,2]. At the global spatial scale, human activities directly affect the change in biodiversity. For example, fertilization changes the composition of local species, and biological invasion breaks the stability of local ecology [3,4]. Biological invasions fundamentally accelerate biodiversity loss, resulting in changes in ecosystems and their functions [5–7]. As one of the most challenging environmental problems of the 21st century, invasion of native ecosystems by harmful plants seriously threatens the global ecological environment and biodiversity [8,9]. *Pedicularis kansuensis* is an annual or biennial root hemiparasite endemic to China and found mainly in Xinjiang, Qinghai, western Sichuan, and Southwest Gansu, and it is rapidly expanding in the alpine grassland of Bayanbulak, Xinjiang, in western China. It has been reported that, similar to other typical invasive native species or locally expanding species [10], this species can spread rapidly in an appropriate ecosystem (grassland in the case of *P. kansuensis*) in a short period of time and shows strong invasiveness [10,11]. The main component species

of the Bayanbulak alpine grassland are the gramineous *Stipa purpurea* and *Festuca ovina* [12], but, since 2000, *P. kansuensis* has appeared in this grassland and exhibited rapid expansion, becoming the dominant species in the Bayanbulak grassland. Studies have shown that the palatability of *P. kansuensis* is poor, and the livestock do not like to eat it; therefore, the dominance of this species has caused great harm to the development of local animal husbandry in this region [13].

According to previous studies, the high seed production and reproductive capacity of *P. kansuensis* meant that this plant is distributed in clusters, which inhibits the growth of local, high-quality herbage, resulting in a rapid decrease in the proportion of good herbage, an increase in inedible herbage, a decline in grassland productivity, and an increasingly serious final contradiction between grass and livestock [14]. Studies have shown that the large-scale occurrence of *P. kansuensis* accelerated the reverse succession of the Bayanbulak grassland, resulting in a vicious cycle within the ecosystem and further increasing the difficulty of managing such degraded grassland [15]. At the same time, the unique root hemiparasitic characteristics of *P. kansuensis* result it in having obvious advantages in competition for nutrition, light, and water with grasses and legumes. Especially in those areas where the grassland is overgrazed and vegetation is seriously damaged, the invasion rate is rapid, making it difficult for other species to grow and develop [16]. Therefore, the effective control of *P. kansuensis* has become the major problem in the restoration of the Bayanbulak grassland. In order to effectively control *Pedicularis* spp., scholars have carried out a lot of research at the early stage of invasion. For example, cutting experiments on *Pedicularis myriophylla* found that cutting could effectively reduce the incidence of *P. myriophylla* over a large area [11]. Shading experiments conducted on John English Prairie in Illinois, USA, found that shading could also effectively inhibit the growth of *Pedicularis canadensis*, and the inhibitory effect increased with increased shading [17]. Houyuan et al. (2011) [18] found that when the leaf extract concentration of *Oxytropis ochrocephala* or *Artemisia nanschanica* reached 100 g L$^{-1}$, application of the extract could directly kill seedlings of *P. kansuensis*. In cultivated grassland in the Yellow River source region, it was found that application of herbicides such as 2,4-D butyl ester, fluoroglycofen-ethyl, and tribenuron-methyl could significantly reduce the height, coverage, and biomass of *P. kansuensis* but had no effect on *Ligularia virgaurea* [19].

Numerous studies have found that appropriate addition of nutrients can promote the quality and yield of plants [20], but, if the amount of fertilizer applied is constantly increased, it can ultimately be counterproductive. Therefore, there is a certain optimal threshold value for nutrient addition [21]. However, up to now, most nutrient addition experiments have focused on achieving high-quality herbage, and there is little research on the impact of nutrients on inhibiting locally invasive plants. In view of this, the research described in this paper considered *P. kansuensis*, a native invasive species in the Bayanbulak alpine grassland, as the research object and conducted a trial of short-term nutrient addition on *P. kansuensis* in the field. Through determination of the height, coverage, richness, biomass, and root/shoot ratio of *P. kansuensis* under four treatments, namely, no added nutrients (control, CK), low-added-nitrogen (N3), high-added-nitrogen (N9), and added phosphorus (P), the changes observed in the *P. kansuensis* population were analyzed. The relationship between different nutrient treatments and the population characteristics of *P. kansuensis* was compared and analyzed.

## 2. Materials and Methods

### 2.1. Site Description

The experiment was located at a fixed sampling site (83°42.5′ E, 42°53.1′ N, 2464 m a.s.l.) at Bayanbulak Grassland Ecosystem Research Station, Xinjiang Institute of Ecological Geography and Geography, Chinese Academy of Sciences. Bayanbulak alpine grassland is located in the northwest of Hejing County, Bayangol Mongolian Autonomous Prefecture, Xinjiang Uygur Autonomous Region, covering a width of 136 km from north to south and a length of 270 km from east to west, with a total area of $2.33 \times 10^6$ hm$^2$. The terrain is

relatively flat but inclines from northwest to southeast and is a typical alpine grassland type, and the climate is windy without an obvious frost-free period. The annual precipitation of the Bayanbulak grassland is 276.2 mm, and the annual evaporation is 1022.9–1247.5 mm. The average annual temperature is $-4.7\,^{\circ}$C, the average annual wind speed is 2.6 m s$^{-1}$, and the sunshine duration is 2466–2616 h [22]. Soil organic carbon content is 38.56 g kg$^{-1}$, total nitrogen content is 4.11 g kg$^{-1}$, total phosphorus content is 0.77 g kg$^{-1}$, total potassium content is 18.37 g kg$^{-1}$. The dominant grassland species in the study area is *P. kansuensis*, with accompanying species such as the grasses *Stipa purpurea*, *Festuca ovina*, and *Koeleria cristata*, the sedge *Carex stenocarpa*, etc. The main soil types are alpine meadow soil and meadow steppe soil. The specific monthly temperature and precipitation data at the experimental site in 2020 and 2021 are presented in Figure 1, courtesy of Bayanbulak Grassland Ecosystem Research Station, Chinese Academy of Sciences.

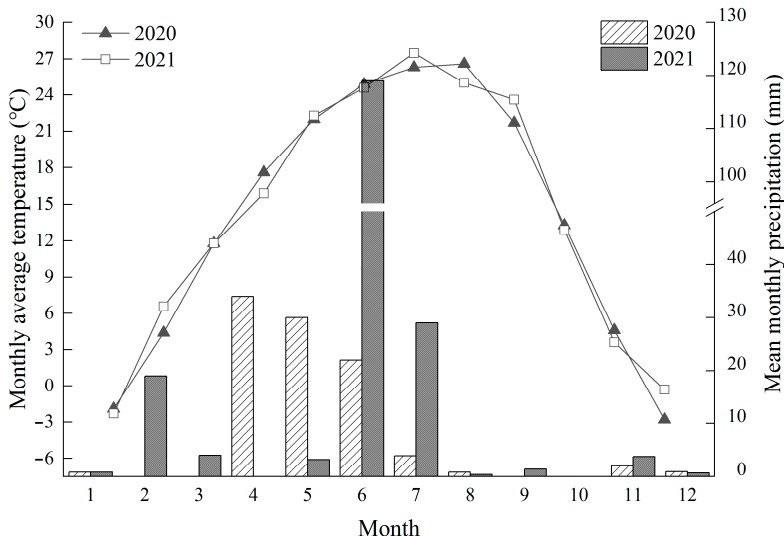

**Figure 1.** Monthly mean precipitation and temperature from 2020–2021 at the experimental site.

## *2.2. Experimental Design*

A large experimental site of 100 × 100 m was set up in a flat area of grassland dominated by *P. kansuensis* near the Bayanbulak Grassland Ecosystem Research Station of the Chinese Academy of Sciences. The nutrient addition experiment was set up in the sample plot using a random block group design. The area of each sample plot was 4 m × 3 m. Nitrogen was supplied as urea ($CH_4N_2O$) for nitrogen, with two treatments, low-nitrogen (N3), expressed as 3 g m$^{-2}$ nitrogen, and high-nitrogen (N9), expressed as 9 g m$^{-2}$ nitrogen; 30 kg ha$^{-1}$ (=3 g m$^{-2}$) is the average nitrogen fertilizer rate used in the arid area of Xinjiang, whereas 90 kg/ha represents the rate in a high-intensity farmland system settlement in North China. Phosphate was supplied as heavy superphosphate ($Ca(H_2PO_4)_2{\cdot}H_2O$), with one treatment, at a dose of 10 g m$^{-2}$; the treatment without nutrient addition was the control (CK). Each group was represented by four replicate plots, making 16 plots in total. Two 1 m × 1 m quadrats were set near the cross point of each sample plot and ten *P. kansuensis* plants were selected as the observation plants in each quadrat; one quadrat was used for data collection in 2020, and the other was used for data collection in 2021, with the buffer zone between each plot being 2 m (Figure 2). The pre-weighed granular fertilizer was mixed with the in situ soil and then evenly scattered in the quadrat. Fertilizer application time in 2020 and 2021 was mid-June and mid-July, respectively.

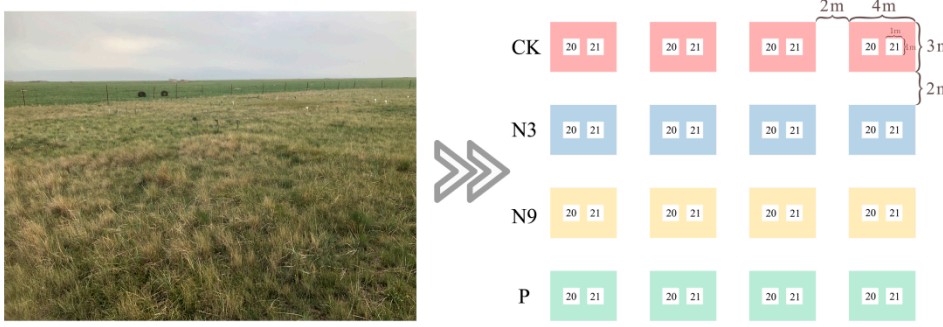

**Figure 2.** Plot layout. In the figure, 20 is the survey quadrat in 2020, and 21 is the survey quadrat in 2021.

*2.3. Investigation and Determination of Plant Samples*

In each quadrangle, the changes in *P. kansuensis* were observed and recorded. Ten plants of *P. kansuensis* were randomly selected and labeled directly in the field. The height, coverage, richness, and biomass of plants in the quadrangle were recorded and measured. The natural height of each plant was measured using a tape measure. The point quadrat method was used to measure the coverage. The 1 $m^2$ quadrat was divided into 100 points, with a needle at each point; each needle was classified as "yes" or "no" with respect to whether the needle was incident or not on *P. kansuensis*, and the coverage was calculated as the percentage of "yes" points. The number of plants in the sample is recorded by the counting method. The aboveground biomass was removed by mowing, put into the labeled envelope bag, brought back to the laboratory, and the aboveground dry weight of the plant was measured after drying in an oven at 65 °C until a constant weight was obtained [23]. The belowground biomass of three soil layers (0–10 cm, 10–20 cm, 20–30 cm) was sampled with an auger (5 cm in diameter) in the center of the sampling square and the core was transferred to gauze mesh bags. After being brought back to the laboratory for cleaning, the roots were dried in an oven at 65 °C to a constant weight and then weighed and recorded as belowground biomass. The root/shoot ratio = belowground biomass/aboveground biomass.

*2.4. Statistical Analyses*

In this experiment, the significance analysis of the height, coverage, density, biomass and root/shoot ratio of *P. kansuensis* was conducted by one-way analysis of variance (ANOVA) in SPSS (PASW Statistics 26; IBM Corporation, Armonk, NY, USA); all variables were tested and shown to approximate to a normal distribution. Two-way ANOVA was used to examine the effects of nutrient addition and time (as the main effects) and their interaction on the height, coverage, density, biomass, and root/shoot ratio of the *P. kansuensis* population. Tukey's honest significant difference (Tukey's HSD) method was used for multiple comparison in different treatments, and for comparisons between two levels the paired *t*-test was used. Pearson's correlation coefficient was used to quantify the relationship among all the variables of the Pedicularis population under different treatments. The significance level was 0.05. The experimental data were processed and analyzed in Excel 2010 (Microsoft Corporation, Albuquerque, NM, USA). Graphics presentation was achieved using Origin 2021 (OriginLab Corporation, Northampton, MA, USA). The summary statistics were expressed as mean ± standard error.

**3. Results**

*3.1. Effects of Nutrient Supplementation on Population Characteristics of P. kansuensis*

There was no same-change trend in plant height under nutrient addition for two consecutive years (Figure 3a). Results showed that the plant height of *P. kansuensis* was significantly increased by 32.21% under the 2020 N3 treatment compared with the control group (*p* < 0.05, Figure 3a). The height of plants under the N9 and P treatments increased

by 17.17% and 17.74% compared with control group, respectively, albeit non-significantly. There was no significant difference in height of *P. kansuensis* among the three treatments. Under the N9 treatment, the height of the plants in the second growing season reached the lowest value, 2.50 ± 0.05 cm, which was significantly lower than that of the control group ($p < 0.05$, Figure 3a). There was no significant difference between the other two treatments compared with the control group. There was a significant difference in height between two treatments (N3 and P) and N9 treatments in 2021 ($p < 0.05$, Figure 3a).

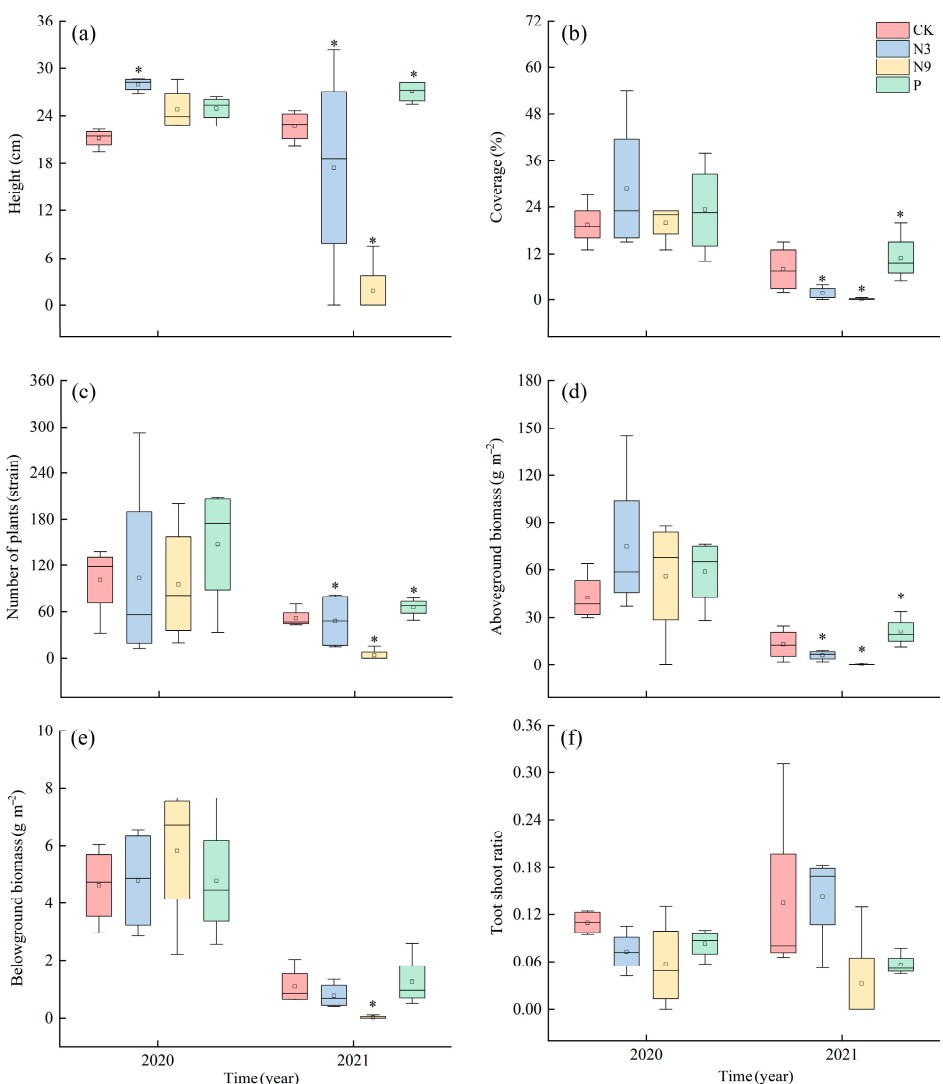

**Figure 3.** Effects of nutrient addition on *P. kansuensis* population characteristics. CK: control, no-nutrient addition; N3: low-nitrogen treatment; N9: high-nitrogen treatment; P: phosphate treatment. (**a–f**) is height, coverage, richness, aboveground biomass, belowground biomass, root/shoot ratio. *: there were significant differences between treatments at the level of $p < 0.05$. The error bars represent a 1.5 range in IQR. The horizontal bars represent the median line. The square represents the mean value.

Compared with the control group, the population coverage of *P. kansuensis* in all nutrient addition treatments increased in the 2020 growing season (Figure 3b). The population coverage of *P. kansuensis* reached the highest value of 28.75 ± 8.97% in the first growing season in response to the N3 treatment. Compared with the control group, N9 and P treatments increased coverage by 2.56% and 19.23%, respectively. There was no significant difference between the control group and any of the three treatment groups. In the 2021 growing season, however, both nitrogen treatments reduced the coverage of *P. kansuensis*.

Under the high-nitrogen treatment, the coverage reached the lowest value of $0.13 \pm 0.02\%$, which was significantly lower than that of the control group ($p < 0.05$, Figure 3b). The *P. kansuensis* plant coverage under the P treatment increased by 37.50% compared with the control group, albeit non-significantly. The coverage of *P. kansuensis* was significantly higher in the P treatment than in either nitrogen treatment group ($p < 0.05$, Figure 3b).

The richness of *P. kansuensis* increased (N3, P) and decreased (N9) in the growing season of the first year in response to the addition of various nutrients (Figure 3c). There was no significant difference between the four different treatments. In the second growing season, both nitrogen treatments reduced the richness of *P. kansuensis*. Among them, compared with the control, N3 treatment decreased by 6.79% though non-significantly. With N9 treatment, the richness of *P. kansuensis* decreased to about three plants, which represented a significant decrease of 92.72% compared with the control group ($p < 0.05$, Figure 3c). Compared with the control group, the richness of the P treatment increased non-significantly by 27.67%. Among the treatments, there were significant differences in richness between N9 and two other treatments (N3, P) ($p < 0.05$, Figure 3c).

The aboveground biomass of *P. kansuensis* was increased by the three groups of nutrient addition treatments in the growing season of the first year (Figure 3d). Under the N3 treatment, the aboveground biomass of *P. kansuensis* reached its highest value, which was $74.81 \pm 23.98$ g m$^{-2}$. Compared with the control group, the aboveground biomass of the groups (N3, N9, P) increased by 75.41%, 31.37%, and 37.88%, respectively, with none of the effects being significant. In the second growing season, in response to the N9 treatment, the aboveground biomass of *P. kansuensis* reached its lowest level at $0.21 \pm 0.04$ g m$^{-2}$, significantly lower than the control group by 98.34% ($p < 0.05$, Figure 3d). Although the N3 treatment reduced aboveground biomass, the effect was not significant. The aboveground biomass under the P treatment increased by 60.92% compared with the control group, albeit non-significantly. In the same year (2021), the aboveground biomass under the N3 and N9 treatments was significantly lower than that under the P treatment ($p < 0.05$, Figure 3d).

In the growing season of the first year, the nutrient addition treatments showed an increase (N3, N9) or a decrease (P) in belowground biomass compared with the control group (Figure 3e). Compared with the control group, the belowground biomass under N3 and N9 treatments was increased by 3.63% and 26.08%, respectively, and the P treatment was increased by 3.41% compared with the control group. There was no significant effect on the belowground biomass of the three groups. In the second growing season, both nitrogen treatments inhibited the belowground biomass of *P. kansuensis*. The belowground biomass under the N3 treatment was 28.54% lower than the control group, with no significant effect. Under the N9 treatment, the belowground biomass of *P. kansuensis* was the lowest, and its value, $0.03 \pm 0.01$ g m$^{-2}$, was significantly different from that of control group ($p < 0.05$, Figure 3e). Compared with the control group, the P treatment increased by 14.61%, although the effect was not significant. There was no significant effect among the three nutrient addition treatments.

As shown in Figure 3f, nutrient addition decreased the root/shoot ratio of *P. kansuensis* population (except for the N3 treatment in the first growing season). Compared with the control group, the nutrient treatments (N3, N9, P) reduced the root/shoot ratio in the middle of the 2020 growing season by 33.92%, 48.48%, and 24.77%, respectively, albeit non-significantly. There was no significant change in this parameter among the different treatments in the same year. In 2021, the N3 treatment increased by 6.52% compared with the control group, but this change was not significant. The root/shoot ratio of the N9 treatment and the P treatment were 75.91% and 58.30% compared with the control group, with no significant effect. There was no significant difference in the root/shoot ratio among the different nutrient treatments in the same year.

### 3.2. Effects of Nutrient Addition and Time Main Effects and Interaction on Population Characteristics of P. kansuensis

The characteristics of *P. kansuensis* population were analyzed by two-way ANOVAs through nutrient addition and time effects and the nutrient × time interaction. Nutrient addition significantly affected the root/shoot ratio of *P. kansuensis* ($p < 0.05$, Table 1) and plant height ($p < 0.01$, Table 1), whereas plant height, coverage, richness, aboveground biomass and belowground biomass of *P. kansuensis* all exhibited significant responses to time ($p < 0.01$, Table 1), and root/shoot ratio had no significant effect (Table 1). There was a significant interaction between nutrient addition and time on the plant height of *P. kansuensis* ($p < 0.01$, Table 1), meaning that the effect of added nutrients on plant height differed significantly between 2020 and 2021.

**Table 1.** Effects of nutrient addition and time on F-values from ANOVA on population characteristics of *P. kansuensis*.

| Index | N | T | N × T |
|---|---|---|---|
| Height | 14.380 ** | 12.995 ** | 14.613 ** |
| Coverage | 0.952 | 33.116 ** | 1.411 |
| Richness | 1.019 | 9.176 ** | 0.190 |
| Aboveground biomass | 0.663 | 31.022 ** | 1.022 |
| Belowground biomass | 0.626 | 65.262 ** | 0.484 |
| Root/shoot ratio | 3.300 * | 0.029 | 0.564 |

Note: N: nutrient addition. T: time. The data in the table indicates F-values. * indicates $p < 0.05$, ** indicates $p < 0.01$.

### 3.3. Relationship between Population Characteristics of P. kansuensis under Different Nutrient Treatments

Data analysis showed that there was a significant positive correlation between the coverage of *P. kansuensis* and aboveground biomass in the first growing season under treatment N3 ($p < 0.05$, Figure 4), and there was no significant relationship between these parameters in the second growing season. In the second growth season under treatment N9, except for the absence of a significant correlation between height and the other parameters, there was a significant positive correlation between the other parameters ($p < 0.05$, Figure 4). There was no correlation between parameters under the N9 treatment in 2020 and the parameters under the P treatment in the same year. Under the P treatment in 2021, the coverage was positively correlated with aboveground biomass, belowground biomass, and root/shoot ratio ($p < 0.05$, Figure 4). There were significant positive correlations between aboveground biomass and both belowground biomass and root/shoot ratio ($p < 0.05$, Figure 4) and a significant positive correlation between belowground biomass and root/shoot ratio ($p < 0.05$, Figure 4). There was no significant relationship between the other parameters.

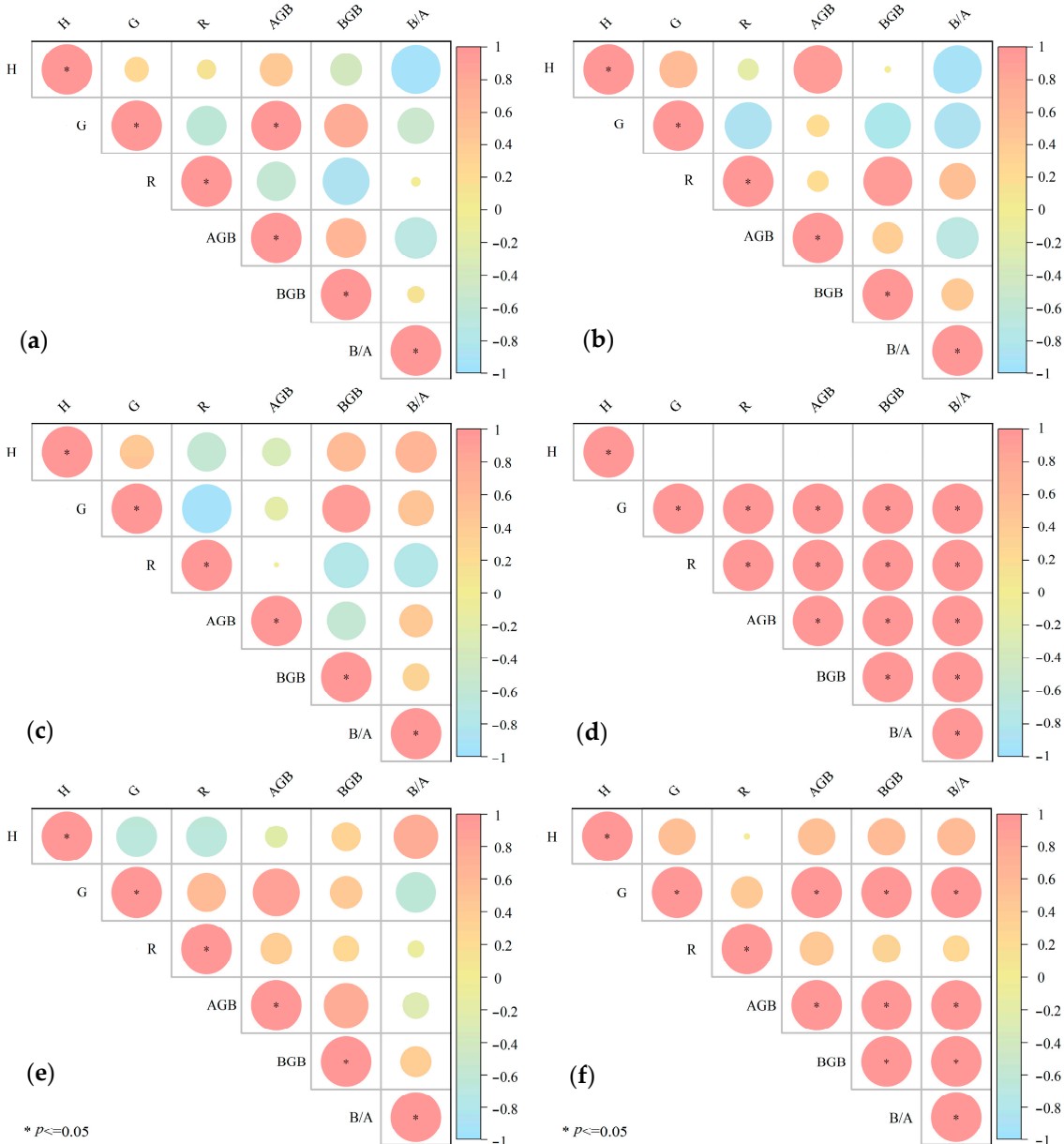

**Figure 4.** Correlation between *P. kansuensis* population characteristics. *, a significant correlation at the *p* < 0.05 level; H, height; G, coverage; R, richness; AGB, aboveground biomass; BGB, belowground biomass; B/A, root/shoot ratio. (**a**–**f**) is 20–N3, 21–N3, 20–N9, 21–N9, 20–P, 21–P, where 20 = year 2020, 21 = year 2021. N3, N9, P refer to nutrient addition treatments. The size of the circle in the diagram is relevant; the greater the circle, the stronger the correlation, and the weaker the opposite.

## 4. Discussion

Successful invasion by plants depends on the characteristics of the plants on the one hand, and on the environmental conditions on the other hand [24,25]. As an indispensable factor in plant growth and development, the effects of different nutrient contents on plant growth and development are also different. Previous studies have found that nitrogen is the nutrient element required most during the growth and development of plants [26,27]. Studies have shown that nutrient addition to other plant species can significantly increase plant height, tiller number, root length, and pod number [28–32]. However, some studies reported that nutrient addition had no significant effect on plant height [33–35]. In the current study, through two consecutive years of nutrient addition experiments, the height,

coverage and richness of *P. kansuensis* increased in the first growing season compared with the control group. However, in the second growing season, both nitrogen treatments reduced the height, coverage and richness of *P. kansuensis,* especially the three groups of indexes (height, coverage, and richness) of *P. kansuensis* decreased significantly under high-nitrogen addition, whereas the height, coverage and richness of *P. kansuensis* were increased under phosphorus treatment in the same year. These results indicated that the growth and development of *P. kansuensis* could be effectively inhibited by addition of a high concentration of nitrogen. This result was inconsistent with previous research results, with the main reason being related to the hemiparasitic characteristics of the root of *P. kansuensis*. As the preferred hosts of *P. kansuensis*, graminaceous plants have the advantage of efficient nitrogen utilization [36]; that is, nitrogen addition could increase the coverage and height of graminaceous plants. When resources are abundant, plants increase their investment in aboveground parts and reduce the distribution of nutrients to belowground roots. Especially when high nitrogen concentrations are added, the increase in the coverage and height of graminaceous plants above ground can directly lead to a disadvantage to *P. kansuensis* with respect to light acquisition [37]. Under high-nitrogen conditions, graminaceous plants reduce their nutrient investment in the roots, which affects *P. kansuensis*, which is a root hemiparasitic plant. As a result, the reduction in host roots increases the difficulty of haustorium initiation by *P. kansuensis*, blocking the nutrient transport and transfer in *P. kansuensis*, and ultimately affecting its growth [38]. In addition, the rapid growth of *P. kansuensis* caused by nitrogen addition in the first growing season may be related to the local rainfall in 2021 being lower than that in 2020. Nutrient addition was mainly carried out during the vegetative growth period of *P. kansuensis*, and the interaction of nutrient and water in 2020 was more conducive to plant nutrient uptake and utilization and hence growth. However, in the second growing season, with the continuing increase in nutrients, a large amount of nitrogen accumulation had a toxic effect on *P. kansuensis* [39], thus inhibiting its growth.

Grassland biomass is a comprehensive reflection of the structure and function of the grassland ecosystem [40]. The addition of nutrients can increase the aboveground biomass and productivity of plants [41–46] but also reduces the belowground biomass [47]. This result was fully demonstrated by nitrogen addition experiments in the Himalayan alpine steppe [48], which affected nutrient allocation. Studies have shown that, in an ecosystem with relatively low nutrients, plants adapt to the environment through coordination and trade-off of investment and allocation between the aboveground part and the belowground part [40]. For example, when nitrogen and phosphorus are deficient, *Pedicularis tricolor* obtains more nutrients through increased investment in the roots [49], mainly due to the fact that haustoria of root semi-parasitic plants are more abundant in lateral roots [50], and increased length of lateral roots may provide more opportunities for the formation of haustoria. However, it was found in the present study that two groups of nitrogen and phosphorus treatments increased the aboveground and belowground biomass of the *P. kansuensis* population in the first year growing season. In the growing season of the second year, however, with the continuous addition of nitrogen, aboveground and belowground biomass showed a gradual decrease, and there was a significant difference between the control group and the high-nitrogen treatment group. However, the two biomass parameters showed an increase under phosphorus treatment. Some studies have proved that there is a trade-off between plant competition for soil resources and with the increase in habitat community productivity; the competition between plants shifts from belowground to aboveground [51]. In the early stage, the increase in biomass of *P. kansuensis* could be related to the higher precipitation in 2020 than in 2021, because the increase in precipitation in the semi-arid grassland increases the concentration of soil nitrogen, which alleviates the restriction of nitrogen availability on plant growth to a certain extent. However, by increasing the allocation of biomass to the root absorption organs, *P. kansuensis* can obtain more soil nutrients to meet its own growth and development needs. However, there was a negative effect between the *P. kansuensis* population vigor and

nitrogen addition. The effect of high nitrogen concentration would reduce the number of *P. kansuensis* plants, reduce its aboveground and belowground biomass, inhibit its growth, or even cause its elimination. This finding was consistent with previous studies, namely that adding sufficient nitrogen could inhibit the growth and reproduction of *P. kansuensis* [37].

As an important parameter for estimating carbon storage in grassland ecosystems, root/shoot ratio is of great importance [52] because changes in root/shoot ratio are closely related to the aboveground and belowground biomass of plants and can reflect the distribution and transfer of resources by plants. Nutrient addition certainly has an impact on root/shoot ratio. However, the conclusions from other studies have been inconsistent. For example, some researchers showed that the application of nitrogen would reduce the root/shoot ratio of plants [53], whereas others found that the application of phosphorus fertilizer had no significant effect on the root/shoot ratio [54]. In the current study, it was found that different concentrations of nitrogen and phosphorus treatments could reduce the root/shoot ratio of *P. kansuensis* populations, a finding which was consistent with the previous research conclusion that the application of nutrients within a certain concentration range could reduce the root/shoot ratio of plants [53]. The main reason lies in nitrogen limitation in the grassland ecosystem. In response to the application of nitrogen fertilizer, and the nitrogen limitation of the ecosystem decreases or even disappears, resulting in more energy being allocated to the aboveground part of the plant for photosynthesis, increasing the productivity of the plant community and thus reducing the root/shoot ratio of the plant population [55,56]. At the same time, it may affect the microenvironment of the *P. kansuensis* population.

**5. Conclusions**

The study of nutrient addition to the Bayanbulak alpine grassland for 2020–2021 showed that both nutrient addition and time had significant effects on the growth of *P. kansuensis*. In the first growing season, nitrogen supplementation increased the height, coverage, richness, and biomass of *P. kansuensis*. However, in the second growing season, these indexes of *P. kansuensis* tended to decrease, especially under high-nitrogen concentration, indicating that the expansion of the *P. kansuensis* population could be significantly inhibited over time following nitrogen addition. In addition, since only 2 years were used for the experiment in this study, the effect of long-term nutrient addition on *P. kansuensis* populations is unclear and can be analyzed in the future through a longer period of time, combined with different precipitation gradients and grazing intensities.

**Author Contributions:** Conceptualization, Y.L. and Y.G.; methodology, Y.L. and Y.G.; software, H.L.; formal analysis, H.L. and F.F.; investigation, Y.L., Y.G. and H.L.; data curation, H.L.; writing—original draft preparation, H.L., Y.L. and Y.G.; writing—review and editing, Y.L., Y.G., K.L. and F.F.; visualization, Y.L. and Y.G.; supervision, Y.L. and K.L.; funding acquisition, Y.L. and Y.G. All authors have read and agreed to the published version of the manuscript.

**Funding:** This research was supported by the "Young Scholars in Western China" Program of the Chinese Academy of Sciences (grant numbers 2019-XBQNXZ-B-002); Nsfc (National Natural Science Foundation of China)–Xinjiang Joint Fund Project (grant no. U1903104); Youth Innovation Promotion Association, Chinese Academy of Sciences (grant no. 2019429).

**Institutional Review Board Statement:** Not applicable.

**Informed Consent Statement:** Not applicable.

**Data Availability Statement:** Not applicable.

**Conflicts of Interest:** The authors declare no conflict of interest.

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
