# Peer review of "Effects of Nutrient Addition on Pedicularis kansuensis Invasion of Alpine Grassland"

_atmosphere, doi:10.3390/atmos14020367_

Round 1
Reviewer 1 Report
Haining et al. present "Effects of Nutrient Addition on Pedicularis kansuensis Invasion of Alpine Grassland". The issue studied here is to study providing information on the future effective control of the expansion of the P.kansuensis population, as well as give data on fundamental studies and propose hypotheses on the theoretical basis for the effects of nutrient addition on invasive plants.
The comparative novelty of this work is to evaluate for the first time indicating that higher nitrogen addition could effectively slow the invasion of the P. kansuensis population
In my opinion, this work is of interest to researchers in the field of nutrient addition in invasive plants but it requires minor revision before it becomes suitable for publication. The authors should consider the following comments to improve their manuscript.
MA1. The Introduction must be improved by incorporating more recent references including global ecological environment and biodiversity and their effects.
MA2. In conclusion, please the contents detailed should be addressed including future scope and applications for a better understanding of both nutrient addition and time had significant effects on the growth.
Reviewer 2 Report
Dear Authors,
I have reviewed the paper "Effects of Nutrient Addition on Pedicularis kansuensis Invasion of Alpine Grassland ". The aims of the paper are germane with Atmosphere topic. The paper is written with a moderate English level. In my opinion the contribution of this paper to the scientific knowledge is low. There are some flaws and I suggest the corrections in the comments in the file attached.

Reviewer 3 Report
Dear Authors,
The submitted manuscript titled „Effects of Nutrient Addition on Pedicularis kansuensis Invasion of Alpine Grassland” contains very valuable results of nutrient supplementation on number and traits of individuals of an aforementionned species. However, I have found some imperfections, chich (in my opinion) should be improved or at least clarified before an eventual publication.
1. Introduction
· My greatest concern refers to used terminology. I am not convinced if term invasive species in appropriate in this case. According to IUCN the Invasive Alien Species (IAS) are animals, plants or other organisms that are introduced into places outside their natural range, negatively impacting native biodiversity, ecosystem services or human well-being. Therefore the therm expansive species might be better for Pedicularis kansuensis occuring in China. Please, be carefull especially, that expansive native species usually have the same (or very similar) ecological and economic consequences as invasive alien species. Nevertheless, if this species has status invasive please add the appropriate literature source.
· In my opinion the characteristic of a study species should be enlarged (especially range of species, habitat affiliation, mode of reproduction, lifespan)
· Please add the specific aims or working hypotheses at the end of chapter
2. Line 113 In my opinion the Figure presenting the experimenta design would be very useful in understanding the methodology of studies.
3. Lines 134 and 191 What do does the richness of Pedicularis kansuensis mean? Is it the number of inividuals?
4. Results
· In my opinion the text describing outcomes presented in graphs is too large. I suggest to shorten it limiting to pointing out the main trends.
5. In Conclusions section I suggest to add the possibile directions of further investigations.
6. Please look into the following publications. Perhaps some of them would be helpful in manuscript improvements:
· Bao et al. 2015. Effects of the hemiparasitic plant Pedicularis kansuensis on plant community structure in a degraded grassland. Ecol. Res. 30(3): 507-515.
· Hu, J.; Li, K.; Deng, C.; Gong, Y.; Liu, Y.; Wang, L. Seed Germination Ecology of Semiparasitic Weed Pedicularis kansuensis in Alpine Grasslands. Plants 2022, 11, 1777. https://doi.org/10.3390/plants11131777
· Dan Wang, Bochao Cui, Susu Duan, Jijun Chen, Hong Fan, Binbin Lu, Jianghua Zheng 2019. Moving north in China: The habitat of Pedicularis kansuensis in the context of climate change. Science of The Total Environment, 697,133979, https://doi.org/10.1016/j.scitotenv.2019.133979.
· Qin, R.; Wei, J.; Ma, L.; Zhang, Z.; She, Y.; Su, H.; Chang, T.; Xie, B.; Li, H.; Wang, W.; Shi, G.; Zhou, H. Effects of Pedicularis kansuensis Expansion on Plant Community Characteristics and Soil Nutrients in an Alpine Grassland. Plants 2022, 11, 1673. https://doi.org/10.3390/plants11131673
Reviewer 4 Report
The abstract should be amended with some exact data. like the amount of nutrients, and correlations. In the introduction section, this part belongs to the discussion or conclusion section:
The relationship between different nutrient treatments and the population characteristics of P. kansuensis was compared and analyzed. This study can provide information on the future effective control of expansion of the P. kansuensis population, as well as providing data on fundamental studies and proposing hypotheses on the theoretical basis for the effects of nutrient addition on invasive plants.
The results section is clear and well-written.
The conclusion section should also be amended with some numerical data.
Please have in mind the broader readership and state your conclusions in a wider manner.
Otherwise, the manuscript is acceptable.
Round 2
Reviewer 2 Report
Dear Authors, with the changes made the article has been implemented
Author Response
Thank you to the reviewer for your comments on our manuscript.
Reviewer 3 Report
Dear Authors,
In my opinion Your manuscript received the sufficient corrections therefore I do not have any further remarks.
Author Response
Thank you for your comments.